# Structural evolution of nitrogenase over 3 billion years

Bruno Cuevas Zuviría[1,2], Franka Detemple[3], Kaustubh Amritkar[1], Amanda K Garcia[1], Lance Seefeldt[4], Oliver Einsle[3], Betül Kaçar[1]*

[1]Department of Bacteriology, University of Wisconsin-Madison, Madison, United States; [2]Centro de Biotecnología y Genómica de Plantas, Universidad Politécnica de Madrid (UPM)—Instituto Nacional de Investigación y Tecnología Agraria y Alimentaria-CSIC (INIA/CSIC), Campus de Montegancedo, Madrid, Spain; [3]Institute of Biochemistry, University of Freiburg, Freiburg, Germany; [4]Department of Chemistry and Biochemistry, Utah State University, Logan, United States

## eLife Assessment

This **valuable** study presents computational analyses of over 5000 predicted extant and ancestral nitrogenase structures. The data analyses are **convincing**, it offers unique insights into the relationship between structural evolution and environmental and biological phenotypes. The data generated in this study provide a vast resource that can serve as a starting point for studies of reconstructed and extant nitrogenases.

**\*For correspondence:**
bkacar@wisc.edu

**Abstract** Previously, we identified the only dinitrogen reduction mechanism known to date as an ancient feature conserved from nitrogenase ancestors, which we directly tested by resurrecting and integrating synthetic ancestral nitrogenases into the genome of *Azotobacter vinelandii* (Garcia et al., 2023), a genetically tractable, nitrogen-fixing model bacterium. Here, we extend this paleomolecular approach to investigate the structural evolution of nitrogenase over billions of years of evolution by combining phylogenetics, ancestral sequence reconstruction, protein crystallography, and deep-learning based predictions. This study reveals that nitrogenase, while maintaining a conserved multi-meric core, evolved novel modular features aligned with major environmental transitions, suggesting that subtle distal changes and transient regulatory adaptations were key to its long-term persistence and to shaping protein evolution over geologic time. The framework established here provides a foundation for identifying structural constraints that governed ancient proteins and for situating their sequences and structures within phylogenetic and environmental contexts across time.

## Introduction

Life on Earth is approximately 4 billion years old—nearly as old as the planet itself. Over this immense timespan, life has continually interacted with its environment, both shaping and responding to planetary change. Because the fossil record is sparse and incomplete for Earth's earliest eras (*Knoll, 2003*), genetic sequences, especially those encoding enzymes central to biogeochemical cycles, offer a powerful molecular archive (*Garcia and Kaçar, 2019*). Advances in evolutionary modeling, ancestral reconstruction, and deep-learning-based structure prediction (e.g., AlphaFold; *Jumper et al., 2021*) now make it possible not only to reconstruct extinct enzymes but also to examine their structural evolution with greater fidelity, providing a new guide for probing ancient biology.

A prime example of an ancient, globally influential enzyme is nitrogenase. Nitrogenase provides access to bioessential nitrogen via the reduction of highly inert atmospheric dinitrogen ($N_2$) to ammonia ($NH_3$) (*Rucker and Kaçar, 2024*). In select microbes called diazotrophs, this pathway is exclusively performed by members of the nitrogenase metalloenzyme family that catalyze the cleavage of the strong N≡N bond. Age-calibrated, phylogenomic analyses of nitrogenase genes (*Parsons et al., 2021*) and nitrogen isotopic signatures preserved in the geologic record (*Stüeken et al., 2015*) both provide evidence for the existence of biological nitrogen fixation more than 3 billion years ago. Thus, as the sole biological means for fixing essential nitrogen, nitrogenases have underlaid the productivity of the biosphere for most of Earth's history (*Falkowski, 1997*; *Navarro-González et al., 2001*; *Sánchez-Baracaldo et al., 2014*).

Despite the antiquity and ecological importance of nitrogenases, nearly nothing is known about how their enzymatic properties have varied in the past (*Rucker and Kaçar, 2024*), particularly in response to ancient, global environmental transitions and the distribution of nitrogenase genes across ecologically diverse diazotrophs (*Sobol et al., 2025*). Notably, nitrogenases are extremely sensitive to oxygen, which degrades the bound metalloclusters required for their enzymatic activity and multimeric structure (*Robson and Postgate, 1980*). Nevertheless, these enzymes, which originated in anaerobic organisms inhabiting an anoxic planet >3.2 billion years ago (*Stüeken et al., 2015*), evolved through the progressive oxygenation of Earth's surface environment (beginning ~2.4–3 billion years ago) (*Lyons et al., 2024*; *Lyons et al., 2014*) and genetic acquisition by aerobic organisms (*Boyd et al., 2015*). Though several regulatory mechanisms and accessory proteins that today shield nitrogenase from oxygen have been identified (*Dixon and Kahn, 2004*; *Robson and Postgate, 1980*; *Schlesier et al., 2016*; *Takimoto et al., 2022*), molecular features associated with oxygen protection adaptation have not been investigated within the nitrogenase enzyme itself. Another outcome of Earth surface oxygenation is the spatiotemporal shift in the environmental bioavailability of redox-sensitive trace metals used by nitrogenases. Specifically, molybdenum (Mo), vanadium (V), and iron (Fe) (*Johnson et al., 2021*; *Moore et al., 2017*) are proposed to have shaped the diversification of the variably metal-dependent Mo-, V-, and Fe-nitrogenase isozymes (*Anbar and Knoll, 2002*). The emergence of novel features within the nitrogenase enzyme has been suggested as necessary for maintaining biological nitrogen fixation amid global marine geochemical shifts (*Cuevas-Zuvíria et al., 2024*), yet little is known about the molecular changes and constraints that accompanied their diversification and evolution across time. This knowledge gap limits our ability to link molecular innovations to environmental transitions in nitrogenase evolution, obscuring the principles behind an enzyme capable of catalyzing one of nature's most challenging reactions.

A window into the evolution of nitrogenase function can be generated by probing the historical protein structure space representative of this ancient enzyme family. In this study, we present a combinatorial approach for reconstructing ancient nitrogenase structures and tracking their evolution across Earth history. We draw from a phylogenetic dataset of extant and inferred ancient nitrogenase proteins to reconstruct more than 3 billion years of nitrogenase structural evolution. Our study combines ancestral sequence reconstruction with massive machine-learning-enabled structure prediction of present-day and ancestral nitrogenases, as well as protein crystallographic investigations of select, functionally characterized ancestors. We thus present the first effort to predict all extant (770) and ancestral (4608) enzyme structures across the nitrogenase evolutionary tree, and interpret key observations across an enzyme's structural history within the context of planetary environmental change.

## Results

### Structural prediction of ancient nitrogenase enzymes across time

We began our structural investigation of nitrogenase evolutionary history by conducting a large-scale structure prediction analysis, encompassing 5378 protein structures, a more than threefold increase compared to available nitrogenase structures in the PDB. We then analyzed our phylogenetic dataset to identify notable structural changes. Our dataset includes representatives of all three known nitrogenase isozymes, which are differentiated by the metal composition of their active-site metallocluster: Nif (encoded by *nifHDK* genes; incorporates an iron-molybdenum cofactor, 'FeMoco'), Vnf (encoded by *vnfHDGK* genes; incorporates an iron-vanadium cofactor, 'FeVco'), and Anf (encoded by *anfHDGK*

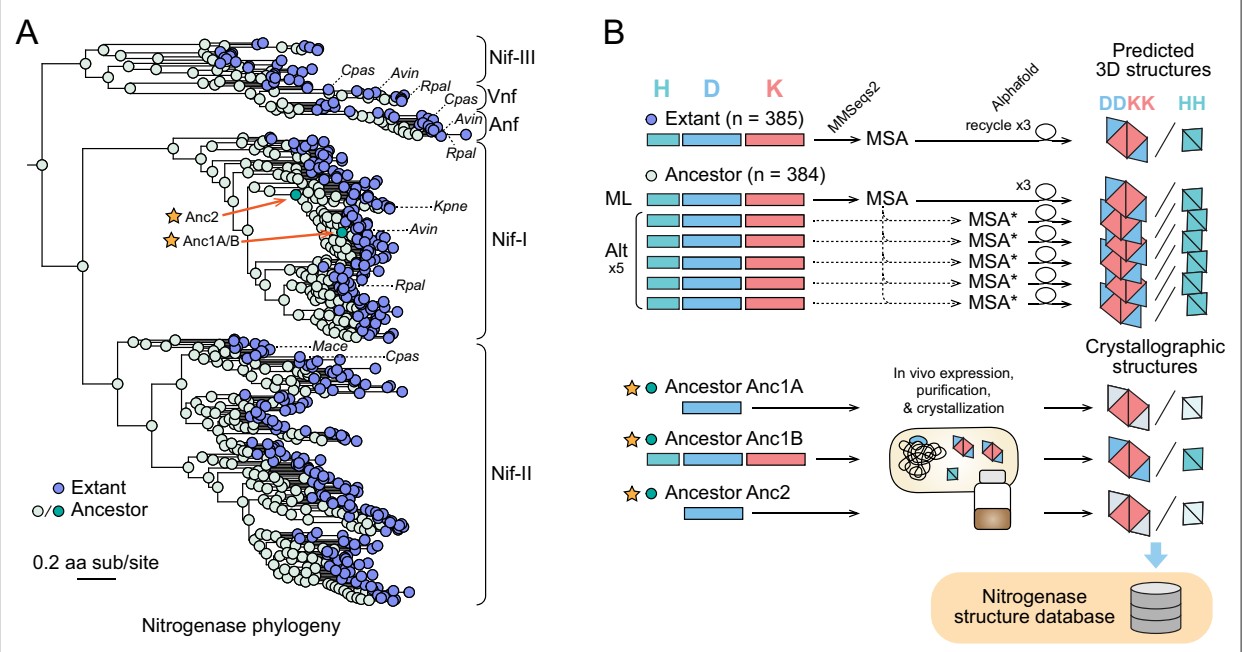

**Figure 1.** Overview of the methodological pipeline for massive protein structure prediction and experimental crystallographic analysis of extant and ancestral nitrogenase enzymes. (**A**) Nitrogenase phylogeny built from concatenated Nif/Vnf/AnfHDK protein sequences. Clades are labeled according to the nomenclature used by *Raymond et al., 2004*. (Nif-I: Group I Nif, Nif-II: Group II Nif, Nif-III: Group III Nif). Nif, Vnf, or Anf homologs from select model organisms are labeled with dashed lines (*Avin: Azotobacter vinelandii*, *Cpas: Clostridium pasteurianum*, *Kpne: Klebsiella pneumoniae*, *Mace: Methanosarcina acetivorans*, *Rpal: Rhodopseudomonas palustris*). Anc1A/B and Anc2 ancestors targeted for crystallographic analysis are labeled with stars. (**B**) Graphical overview of the pipeline for nitrogenase protein structure prediction and crystallization (see 'Materials and methods' for further details). Colored rectangles correspond to protein sequences for H, D, and K subunits. For each ancestral node, protein structures were predicted for the most likely ancestral sequence ('ML') and five alternative sequences ('Alt') reconstructed based on the site-wise posterior probability distributions in the ancestral sequence. Ancestors Anc1A and Anc2 hybrid enzymes were crystallized containing an ancestral NifD subunit and WT NifH and NifK subunits (WT subunits indicated by lighter color). All predicted structures are publicly available at https://nsdb.bact.wisc.edu.

The online version of this article includes the following figure supplement(s) for figure 1:

**Figure supplement 1.** AlphaFold prediction confidence for extant and ancestral nitrogenases.

**Figure supplement 2.** Expanded nitrogenase protein sequence phylogeny (adapted from *Garcia et al., 2020*).

genes; incorporates an iron-only cofactor, 'FeFeco'). H-, D-, and K-subunit sequences were analyzed for each nitrogenase homolog (G-subunit sequences unique to Vnf and Anf were not included in the analysis; see 'Materials and methods'). In all nitrogenase isozymes, H-subunits form a homodimeric reductase component ('HH') that transiently associates with and transfers electrons to a catalytic core (*Einsle and Rees, 2020*). This catalytic component is a heterotetramer composed of two D- and two K-subunits ('DDKK' or $\alpha_2\beta_2$) and contains the active site metallocluster where $N_2$ is bound and reduced to $NH_3$.

The topology of the nitrogenase phylogenetic dataset *Garcia et al., 2023* used in the present study reveals five major clades: Nif-I, Nif-II, Nif-III, Vnf, and Anf (*Figure 1A*). Both Vnf and Anf homologs are nested within the Nif-III clade. Aside from sequence-level homology of their constituent proteins, clades are also distinguished by associated features including metal dependence of the nitrogenase enzyme, taxonomy, host ecology, and complexity of the nitrogenase-associated gene network (*Cuevas-Zuviría et al., 2024*). In extant diazotrophs, nitrogenases require a suite of associated genes for their regulation, assembly, and maintenance (*Martin Del Campo et al., 2022*). The content and complexity of this cellular network for nitrogen fixation tend to be characteristic of the major clades and host ecologies of nitrogenases. For example, Nif-I nitrogenases are primarily hosted by aerobic or facultatively anaerobic diazotrophs that have among the largest number of nitrogenase-associated genes (*Boyd et al., 2015*). By comparison, Nif-II and Nif-III nitrogenases are hosted nearly exclusively by anaerobic diazotrophs and have comparatively smaller nitrogen fixation gene networks (*Cuevas-Zuviría et al., 2024*; *Garcia et al., 2020*). Vnf and Anf isozymes, though sharing certain

associated genes with Nif, are also supported by distinct, dedicated gene clusters (*Garcia et al., 2022*; *Garcia et al., 2020*). We hypothesized that these differences would manifest in correlated structural differences across the enzyme family.

We employed a high-throughput, AlphaFold-based computational pipeline to generate multimeric HH and DDKK structural predictions for each nitrogenase homolog in our dataset (*Figure 1B*). For each of the 385 extant targets, a single structure was predicted. For each of the 384 ancestral targets, six structures were predicted, sourced from the single most likely ancestral protein sequence and five alternative ancestral sequences generated by substitutions at ambiguously reconstructed sites (see 'Materials and methods'). In total, 2689 unique extant and ancestral nitrogenase variants were targeted. All structures were generated in approximately 805 hours, including GPU computations and MMseqs2 alignments performed using two different protocols: one for extant or most likely ancestral sequences, and another for ancestral variants. HH and DDKK structures were predicted independently, resulting in a total of 5378 individual nitrogenase protein structures generated from our pipeline. For comparison, the Protein Data Bank currently contains structures of nitrogenase homologs (from seven different organisms, spanning 15 HH and 24 DDKK 104 structures) (*Supplementary file 1*).

We found that all nitrogenase structures were predicted with high confidence, assessed by the predicted Local Distance Difference Test (pLDDT) (*Jumper et al., 2021*; *Mariani et al., 2013*). pLDDT scores ranged between ~82 and 98 (*Figure 1—figure supplement 1*). Within this range, confidence was lowest for H-subunit structures (median pLDDT ≈ 88) and highest for K-subunit structures (median pLDDT ≈ 98). Furthermore, we built a public web database, 'Nitrogenase Structural DB' (accessible at https://nsdb.bact.wisc.edu), to host all nitrogenase structures predicted in this study. Structures are visualizable, downloadable (PDB format files), and can be searched by sequence or node ID (annotated on the nitrogenase phylogeny in *Figure 1—figure supplement 2*), NCBI taxon ID, and species name.

## Ancient structural insertion events distinguish major nitrogenase clades

We observe that predicted nitrogenase structures have sufficient variations to resolve phylogenetic relationships. To assess structural variation, we calculated the root mean squared deviation (RMSD) using US-Align (*Zhang et al., 2022*). Although in this case, this metric indicates overall small differences (<3 Å) (*Figure 2A*), it still strongly correlates with sequence identity (*Figure 2B*), indicating that even minor structural variations can recapitulate sequence-based phylogenetic distinctions. Hierarchical clustering of nitrogenase structures by pairwise RMSD largely reproduces the major clades (Nif-I, Nif-II, Nif-III, Vnf/Anf) (*Figure 2C*). To confirm that this correlation between sequence identity and structure is not simply an artifact resulting from alignment length bias, we applied two complementary metrics: $RMSD_{100}$ (*Carugo and Pongor, 2001*) and TM-score (*Zhang and Skolnick, 2004*). $RMSD_{100}$ is a normalized variant of RMSD designed specifically to correct structural alignment length bias. The results obtained with these alternative metrics (*Figure 2—figure supplement 1*) were similar to the results of *Figure 2*, differing primarily by a scaling factor of approximately one-third.

TM-score enables a global comparison of structural similarity ranging between 1 (identical) and 0 (no similarity). All DDKK structures have relatively high pairwise TM-scores (>0.84); TM-score >0.5 indicates comparable folding (*Xu and Zhang, 2010*) and hierarchical clustering of nitrogenase structures by pairwise TM-scores largely reproduces the major clades (Nif-I, Nif-II, Nif-III, Vnf/Anf) observed in the protein sequence phylogeny (*Figure 2—figure supplement 2*).

Upon finding a correlation between structure similarity and phylogenetic relationships, we developed a method to sample the inherent uncertainty of ASR outputs by generating sequences with alternative residues at ambiguous positions (see 'Materials and methods'). We predicted the resulting structures and compared the variability of these outputs against the maximum-likelihood sequence. *Figure 2—figure supplement 3* shows the overall distribution of RMSDs for randomized variants against the maximum likelihood variant. RMSDs for most divergent variants fall within two angstroms, indicating very high structural similarity. As shown, the RMSD does not correlate with the distance to the tree's root, indicating no systemic relationship between structural similarity and tree depth (*Figure 2—figure supplement 3B*). Inferred structures, even those derived from deep nodes within the phylogenetic tree, are robust against the statistical uncertainty of ancestral sequence inference. Finally, we computed the all-vs-all RMSD at the DDKK active sites (FeMoco and 8Fe7S; *Figure 2—figure supplement 4A and B*). The results indicate that the active sites have high structural conservation,

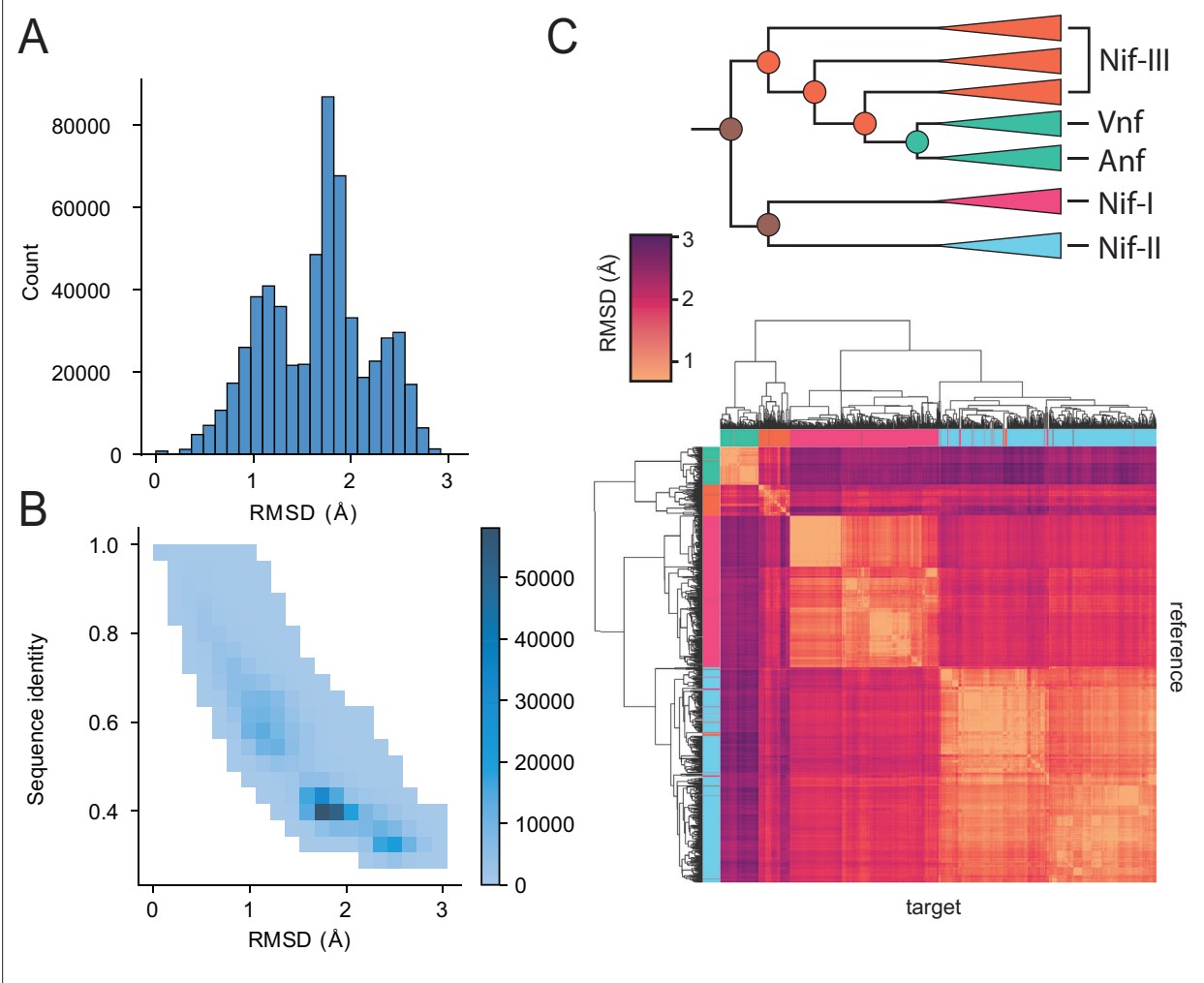

**Figure 2.** Global analyses of nitrogenase DDKK sequence and structural diversity. (**A**) Root mean squared deviation (RMSD) distribution on paired nitrogenase alignments across extant and ML-ancestor pairs. (**B**) Sequence identity and structural similarity (quantified by root mean square deviation (RMSD) of aligned predicted structures) distribution of paired nitrogenase alignments. (**C**) Hierarchical clustering of predicted nitrogenase structures based on structural similarity (RMSD). Each tile in the heatmap corresponds to the RMSD between two nitrogenase structures.

The online version of this article includes the following source data, source code, and figure supplement(s) for figure 2:

**Source code 1.** Global analysis of nitrogenase DDKK sequence and structural diversity.

**Source data 1.** Pairwise alignment distances between DDKK tetramers after alignment with US-align.

**Figure supplement 1.** DDKK $RMSD_{100}$ relationships.

**Figure supplement 2.** DDKK TM-score relationships.

**Figure supplement 3.** Structural variability among sequence variants of ancestral nodes.

**Figure supplement 4.** Residues surrounding metal cofactors and RMSD analysis in nitrogenase.

especially around the 8Fe7S cluster (RMSD <0.5 Å, *Figure 2—figure supplement 4C*), which could be expected given the tight coordination of the metal cofactor with its surrounding cysteines. It is worth noting that AlphaFold2 does not use information about metal cofactors when building the model.

Our global comparisons of predicted nitrogenase structures suggest that, despite relatively high conservation, certain structural distinctions between major clades evolved early and persisted in the extant descendants. We examined our dataset to identify these distinctions and map them to the evolutionary trajectory on the nitrogenase family history.

The structural predictions identified three insertion events that distinguish nitrogenase variants. These insertions likely contribute significantly to the phylogenetic signal encoded within structural

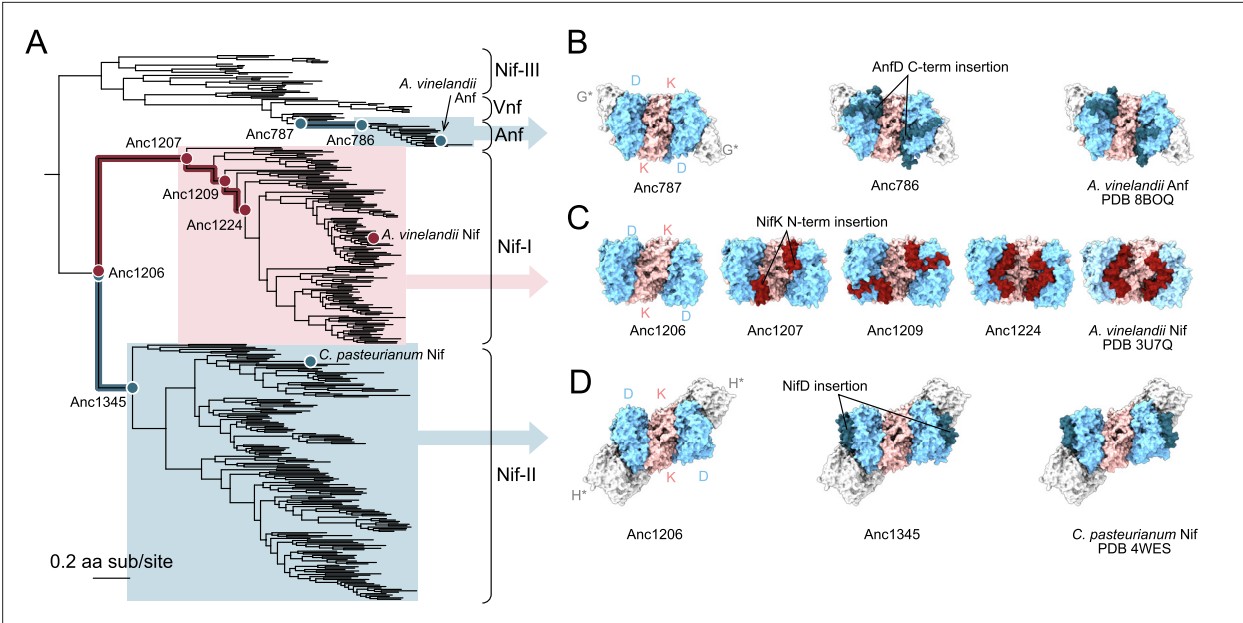

**Figure 3.** Nitrogenase structure variation in a phylogenetic context. (**A**) Nitrogenase protein phylogeny. Branches and ancestral nodes corresponding to structural insertion events, as well as representative extant variants conserving those insertions, are highlighted and/or labeled. Clade and node colors correspond to the subunit for which an insertion is observed (i.e., blue for the D subunit, red for the K subunit). (**B**) Elongation of the NifD C-terminus coincident with the origin of the Anf clade. (**C**) Progressive elongation of the NifK N-terminus through the early evolution of the Nif-I clade. (**D**) Insertion within NifD coincident with the origin of the Nif-II clade. (**B–D**) All visualized structures are predicted unless otherwise specified with the corresponding Protein Data Bank identifier. Bound G- and H-subunit structures were not predicted together with the NifDK structures and are thus indicated with an asterisk. The binding positions of the G- and H-subunit structures are inferred based on alignment with PDB 8BOQ (**Trncik et al., 2023**) and PDB 1M34 (**Schmid et al., 2002**), respectively.

features (**Figure 3A**): a D-subunit C-terminus extension (~60 residues) unique to Anf (**Figure 3B**), a K-subunit N-terminus extension (~8–40 residues) unique to Nif-I (**Figure 3C**), and a D-subunit insertion (~55 residues) unique to Nif-II (**Figure 3D**). The relationships between these insertion events and the major nitrogenase clades are also associated with the significant differences in host ecology and nitrogen fixation genetics across clades.

We found that the K-subunit N-terminus is predominantly found in oxygen-tolerant, aerobic, or facultatively anaerobic diazotrophs, whereas the D-subunit insertion of Nif-II is mostly found in anaerobic diazotrophs. Our ancestral reconstructions constrain the initial appearance of each insertion to the branch leading to the last common ancestor of the clade. Both D-subunit insertions appear within a single branch and thereafter remain relatively conserved in length. By contrast, we observed that the K-subunit N-terminus insertion evolved progressively in length (**Figure 3B and D**) through several deep-time ancestors of the Nif-I clade (**Figure 3C**). The elongation of this insertion recapitulates trends in the K-subunit N-terminus lengths of extant Nif-I homologs, where early-diverged lineages have short (~8 residue) extensions and later-diverged lineages have longer (~40 residue) extensions. The extent to which the N-terminus of the K-subunit affects functional importance in relation to oxygen demand warrants further empirical investigation. As a side observation, our analysis detected no major deletion events in the evolutionary history of nitrogenase. Oldest ancestors (including the last common ancestor of the entire nitrogenase family) and Group III nitrogenases are among the shortest in sequence length and do not contain any of the insertions described above.

The locations of these insertions within the nitrogenase structure and their encoding genes provide insight into their possible functional contributions. First, all insertions are exposed at the surface of the DDKK protein complex (**Figure 3B–D**), and all comprise varying proportions of helix or loop regions in both extant and ancestral structures. The position of the D-subunit C-terminus extension (**Figure 3B**) and the K-subunit N-terminus extension (**Figure 3C**) is similar. Both line the edges of the interface between D- and K- subunit proteins, possibly stabilizing the complex. It is possible that the N-terminus of NifK wraps around the entry point of small molecules (**Barney et al., 2009**; **Morrison et al., 2015**)

and helps in reducing access for oxygen in Nif-I hosted by aerobes. The D-subunit insertion in Nif-II instead protrudes near the inferred HH binding site. Though no crystallographic structures of a bound HH-DDKK nitrogenase complex are available for Nif-II homologs (and we did not predict bound structures here), the Nif-II D-subunit insertion is likely capable of H-subunit interactions based on alignment with the binding site in Nif-I nitrogenases (*Figure 3D*).

## Phylogenetic trends of nitrogenase DDKK complex structures

Following the identification of structural innovations in the history of nitrogenases, we investigated evolutionary trends in calculated structural attributes for DDKK structures, including surface properties and intermolecular contacts (*Figure 4—figure supplement 1A*). Broadly, we found that many structural features carry clade-specific trends and/or are associated with the presence of the insertions described above. For example, total surface area (summed over each protein subunit in the DDKK complex) is distinct between Nif-I, Nif-II, and Nif-III/Vnf/Anf, and greatest in the Nif-II clade (*Figure 4—figure supplement 1B*). Trends in surface area are likely driven in part by the presence and conformation of the surface-exposed insertions. We also examined attributes related to protein-protein interactions within DDKK. In all cases, we found that trends in these attributes were influenced most strongly by the presence of the K-subunit N-terminus extension in the Nif-I clade. Nif-I homologs (and particularly those with longer K-subunit extensions) have the largest number of intersubunit molecular contacts and intersubunit binding affinities, as calculated with Prodigy (*Vangone and Bonvin, 2015*; *Figure 4—figure supplement 1C*, *Figure 4B*). These patterns support our hypothesis that the K-subunit N-terminal extension (*Figures 3C and 4C*) contributes positively to DDKK complex interaction strength and stability. By contrast, homologs with the least number of intersubunit contacts and weakest binding affinities are Nif-III nitrogenases, which notably contain none of the described insertions. We also observed an increase in the total surface area of the nitrogenase complex within the Nif-II group (*Figure 4—figure supplement 1B*). The only structural attribute that we found correlated with ancestor age was the proportion of charged residues across DDKK proteins (*Figure 4—figure supplement 1E*). Accordingly, we observed a weak negative correlation between phylogenetic distance from the root and proportion of charged residues (Spearman R ≈ –0.261). We did not detect similar trends for other attributes. Rather, the attributes of nitrogenase ancestors largely resemble those of their descendants in a clade-specific manner. These results reinforce the idea that major structural distinctions were set early in nitrogenase history and conserved in respective clades.

## Crystalized nitrogenase ancestors confirm evolutionary substitutions at the HH-DDKK interface

In this work, we created a large dataset of nitrogenase structures, expanding upon the limited structural reference data that the prediction methods rely on. Therefore, we continued our structural investigation of nitrogenase evolutionary history by crystallographic characterization of selected nitrogenase ancestors that helped us assess the accuracy of the structure prediction methods. Three previously characterized ancestral variants were targeted from phylogenetic nodes within the evolutionary lineage of *Azotobacter vinelandii* Nif (wild-type, 'WT') (*Figure 5A*; *Garcia et al., 2023*): Anc1A, Anc1B, and Anc2. Anc1A and Anc1B are the same age, whereas Anc2 is reconstructed from an older phylogenetic node. WT, Anc1, and Anc2 belong to a Mo-nitrogenase clade, formerly designated as 'Group I' by *Raymond et al., 2004*. This clade includes homologs from various aerobic and facultatively anaerobic organisms, such as proteobacteria and cyanobacteria (*Garcia et al., 2023*). An estimated maximum age of approximately 2.5 billion years for Group I nitrogenases—and consequently for Anc1 and Anc2—is inferred from the timing of the GOE (*Garcia et al., 2023*; *Lyons et al., 2014*). As an additional distinction, Anc1A and Anc2 contain an ancestral D-subunit protein complexed with WT H- and K-subunit proteins. Anc1B, by contrast, is composed of ancestral H-, D-, and K-subunit proteins. Ancestral variants preferentially bind to the ATP cofactor (*Harris et al., 2024*) and exhibit the same general mechanism for $N_2$ binding and reduction conserved across all known nitrogenases to date, but with decreased activity for reduction of $N_2$ or other substrates (*Garcia et al., 2023*). Thus, the sequence differences between ancestors and WT are sufficient to generate phenotypic changes.

Ancestral nitrogenase DDKK proteins were isolated from genomically modified strains of *A. vinelandii* (*Garcia et al., 2023*), purified by chromatographic methods, and crystallized under anoxic conditions (*Figure 5—figure supplement 1*; see 'Materials and methods'). We found that ancestral

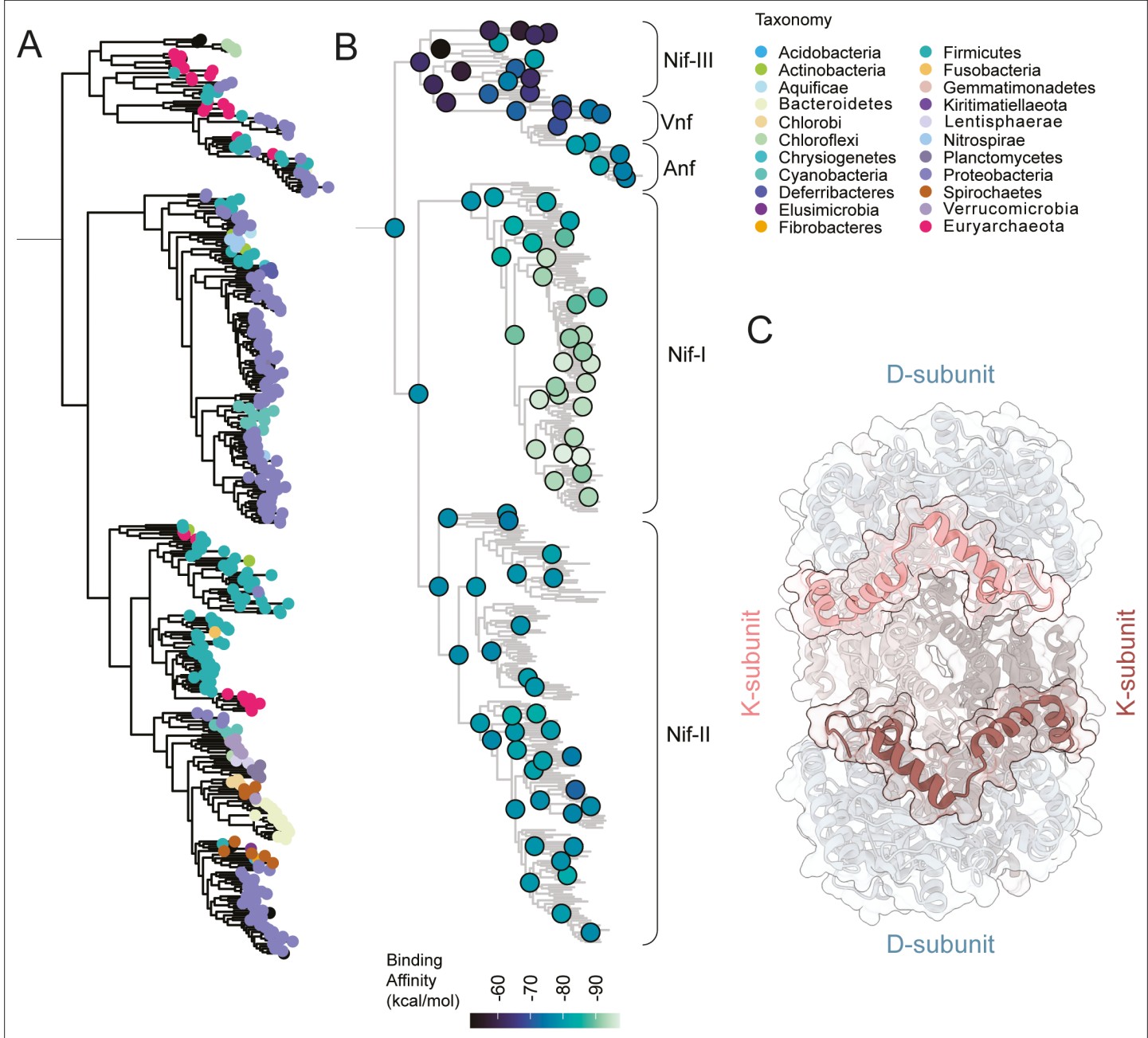

**Figure 4.** Phylogenetic patterns of nitrogenase structural attributes calculated across DDKK proteins. (**A**) Node taxonomy overview. (**B**) Binding affinity prediction for extant and ancestral nitrogenase nodes mapped to the nitrogenase phylogeny. (**C**) Structure overview of the D and K subunit interactions around the N-terminal insertion in Nif-I nitrogenases. Structure corresponding to *Azotobacter vinelandii* NifDK (PDB code: 3U7Q). Note: We reduced the number of displayed phylogenetic nodes to mitigate visual overcrowding; refer to *Figure 4—figure supplement 1* for a complete visualization of all nodes.

The online version of this article includes the following source data, source code, and figure supplement(s) for figure 4:

**Source code 1.** Phylogenetic tree representation of sequence relationships (provided as source data).

**Source code 2.** Source code for phylogenetic tree representation of nitrogenase attributes.

**Source data 1.** Nitrogenase phylogenetic tree.

**Source data 2.** Phylogenetic tree annotations.

**Source data 3.** Nitrogenase DDKK structural data.

**Figure supplement 1.** Phylogenetic mapping of structural features in nitrogenase DDKK proteins.

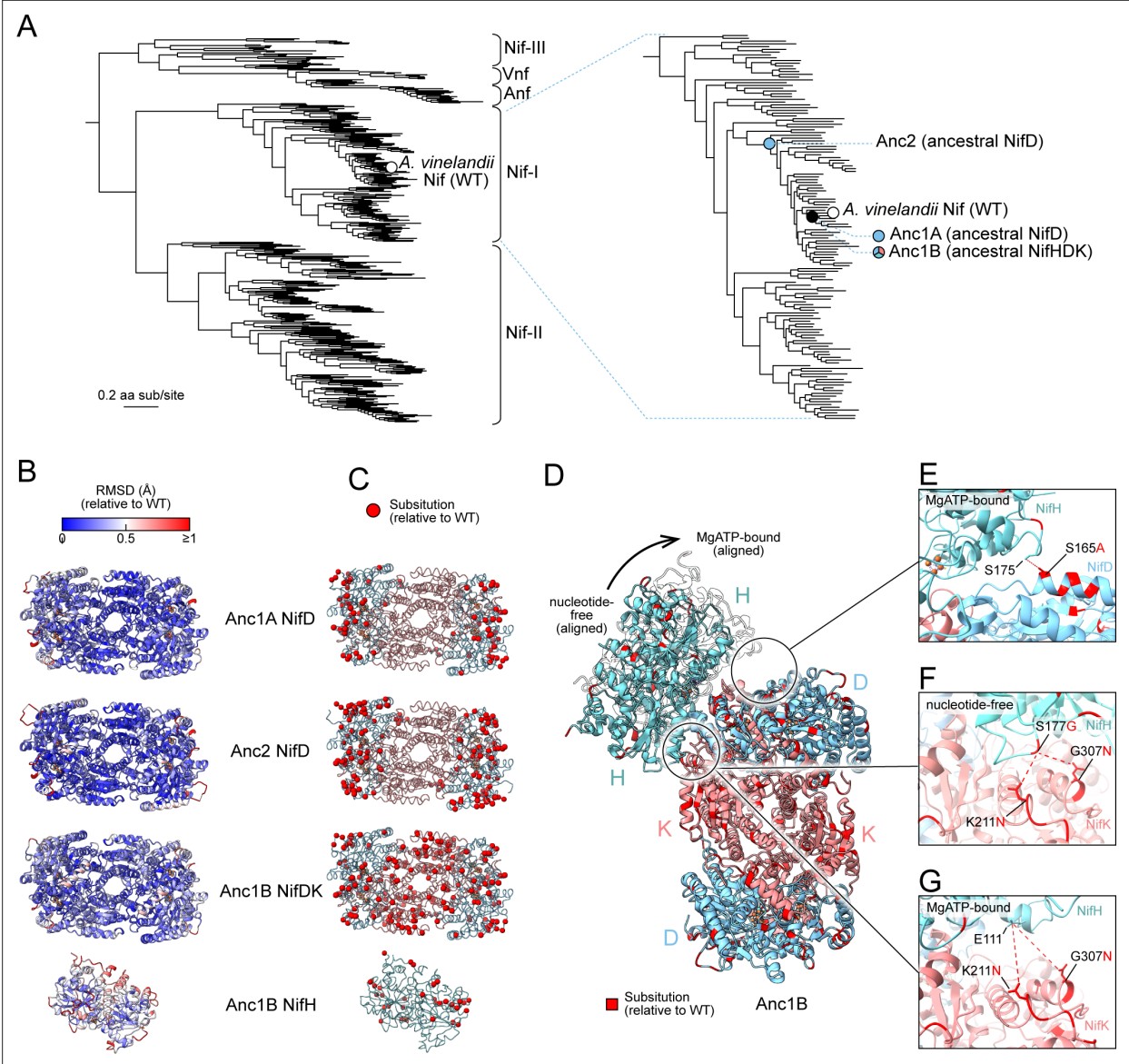

**Figure 5.** Crystal structures of targeted ancestral nitrogenases. (**A**) Nitrogenase protein phylogeny from which nitrogenase ancestral proteins were reconstructed and crystallized for structural characterization. Major clades are labeled following *Garcia et al., 2023*. (**B**) Residue-level root mean squared deviation (RMSD) between the crystallized ancestral and wild-type (*A. vinelandii*) NifDK/NifH structures. (**C**) Spatial distribution of ancestral amino acid substitutions relative to WT. Crystallized protein complexes for Anc1A and Anc2 contain ancestral NifD and WT NifH and NifK. Therefore, ancestral substitutions are only in the NifD subunit for these structures. (**D**) Ancestral amino acid substitutions within the NifH-NifDK interface of Anc1B. Bound positions of NifH are inferred by alignment with either the nucleotide-free (PDB 2AFH; *Tezcan et al., 2005*) or MgATP-bound (PDB 7UT8; *Rutledge et al., 2022*) structures. (**E–G**) Close views of specific, ancestral amino acid substitutions that are inferred to impact NifH-NifDK interactions.

The online version of this article includes the following source data and figure supplement(s) for figure 5:

**Figure supplement 1.** SDS-PAGE analysis of purified proteins.

**Figure supplement 1—source data 1.** Raw, uncropped image for *Figure 5—figure supplement 1*.

**Figure supplement 1—source data 2.** Uncropped, labelled image for *Figure 5—figure supplement 1*.

**Figure supplement 2.** Twofold symmetry axis shift between Anc1B and WT.

**Figure supplement 3.** Sequence alignments between Anc1a, Anc1B, Anc2 and WT *A. vinelandii*.

nitrogenases required different conditions for crystallization relative to that required for modern proteins. Specifically, all ancestors crystallized at pH 6.5, which is unusually acidic for nitrogenase (**Peters et al., 1997**; **Spatzal et al., 2011**). In addition, all ancestral proteins crystallized in space group $P 2_1 2_1 2_1$, which differs from the space groups for the same proteins in *A. vinelandii* NifDK (i.e., WT; space group $P 2_1$), *Clostridium pasteurianum* Nif (space group $P 2_1$), and *Klebsiella pneumoniae* NifDK (space group $C 2$). We observed one ancestral DDKK protein per asymmetric unit. All three ancestors showed anisotropic diffraction: 2.65–2.93 Å for Anc1A, 1.82–2.8 Å for Anc1B, and 1.82–2.42 Å for Anc2. All structures were solved by molecular replacement using WT *A. vinelandii* (PDB 3U7Q).

As in extant nitrogenase proteins, each ancestral D- or K-subunit structure exhibits three Rossmann-fold domains (**Figure 5B**). The P-cluster, an 8Fe7S metallocluster that mediates electron flow to the active site, is located on the pseudo-twofold axis relating the structurally similar D- and K-subunits and was modeled in the all-ferrous $P^N$-state. Both the P-cluster and the active-site cofactor, FeMoco, were modeled with full occupancy. Global structural differences between the ancestors and WT *A. vinelandii* DDKK are relatively minor and within the range of variation for available extant structures. Among all ancestors, RMSDs for all atom positions are largest between Anc1B and WT DDKK (0.36 Å; compared to RMSD of 0.50 Å between *A. vinelandii* and *K. pneumoniae* DDKK). This observation may be explained by the larger number of amino acid substitutions between the fully ancestral Anc1B complex and WT, whereas only the D-subunit contains substitutions in the Anc1A and Anc2 hybrid complexes (**Figure 5C**). Among the two hybrid ancestors, the RMSD between Anc2 and WT (0.24 Å) is smaller than that between Anc1A and WT (0.31 Å). Furthermore, we observe that the position of the twofold symmetry axis relating the two DK heterodimers of a single MoFe protein shifts slightly between Anc1B and WT (**Figure 5—figure supplement 2**). The same was previously observed in VFe protein (**Sippel and Einsle, 2017**) and FeFe protein (**Trncik et al., 2023**) and may reflect slight differences in the interface between the two copies of NifK.

In our previous study, we found that some of the ancestral nitrogenases had decreased $N_2$-reduction activity in vitro relative to WT (**Garcia et al., 2023**). The generation of crystallographic structures for these ancestors provides an opportunity to examine any structural features that might be responsible for the decrease in activity in these variants. Given that most ancestral DDKK substitutions were not close to metalloclusters, we hypothesized that those situated at the DDKK-HH interface might have modified the activity of the ancestors by impeding protein–protein interactions that are necessary for electron transfer to the nitrogenase active site, as well as cofactor loading and complex maturation (**Burén et al., 2020**). Snapshots of crystal structure interface, highlighting mutations and residue interactions are shown in **Figure 5E–G**.

Notably, a non-conserved Nif-D α-helix (positions 165–174 in Anc1B) lies within interacting distance of HH. This helix contains several substitutions across all ancestors, albeit the positioning of polar amino acids differs among Anc1A, Anc1B, and Anc2. Herein, the D-subunit S165A substitution is consistently present across all ancestors and may influence interactions with HH S175 in the ATP-bound complex, potentially reducing stability (**Figure 5E**). Additionally, two unique single-point substitutions in the ancestral K-subunit—K211N and G307N, specific to Anc1B—are likely to impact interactions with HH E111 in the ATP-bound state (**Figure 5G**). Positioned centrally within the DDKK-HH interface, these substitutions alter the electrostatic surface of HH. In Anc1B, the substitution HH S177G is located near the interface with NifDK, with G307N in Anc1B NifK identified as a potential interaction partner (**Figure 5F**).

Predictions for nitrogenase variants had high similarity (average RMSD <1.4 Å) to experimentally determined structures (**Figure 6A**). This was also true for the ancestral nodes selected for crystallographic analysis in this study (**Figure 6B and C**). Intriguingly, none of which were included in the structural database used for AlphaFold training. Beyond the limited number of substitutions within the interfaces of Anc1A/B and Anc2, the changes in the ancestral enzymes were distributed quite evenly across the protein subunits.

## Discussion

In this study, we combined targeted crystallization experiments of ancestral nitrogenase enzymes with large-scale 3D structure predictions spanning both present-day and ancestral nitrogenase diversity. Building on the recent expansion of predicted structures for extant proteins, this work marks a pioneering effort to apply these methods to explore the ancestry of an entire enzyme family. The

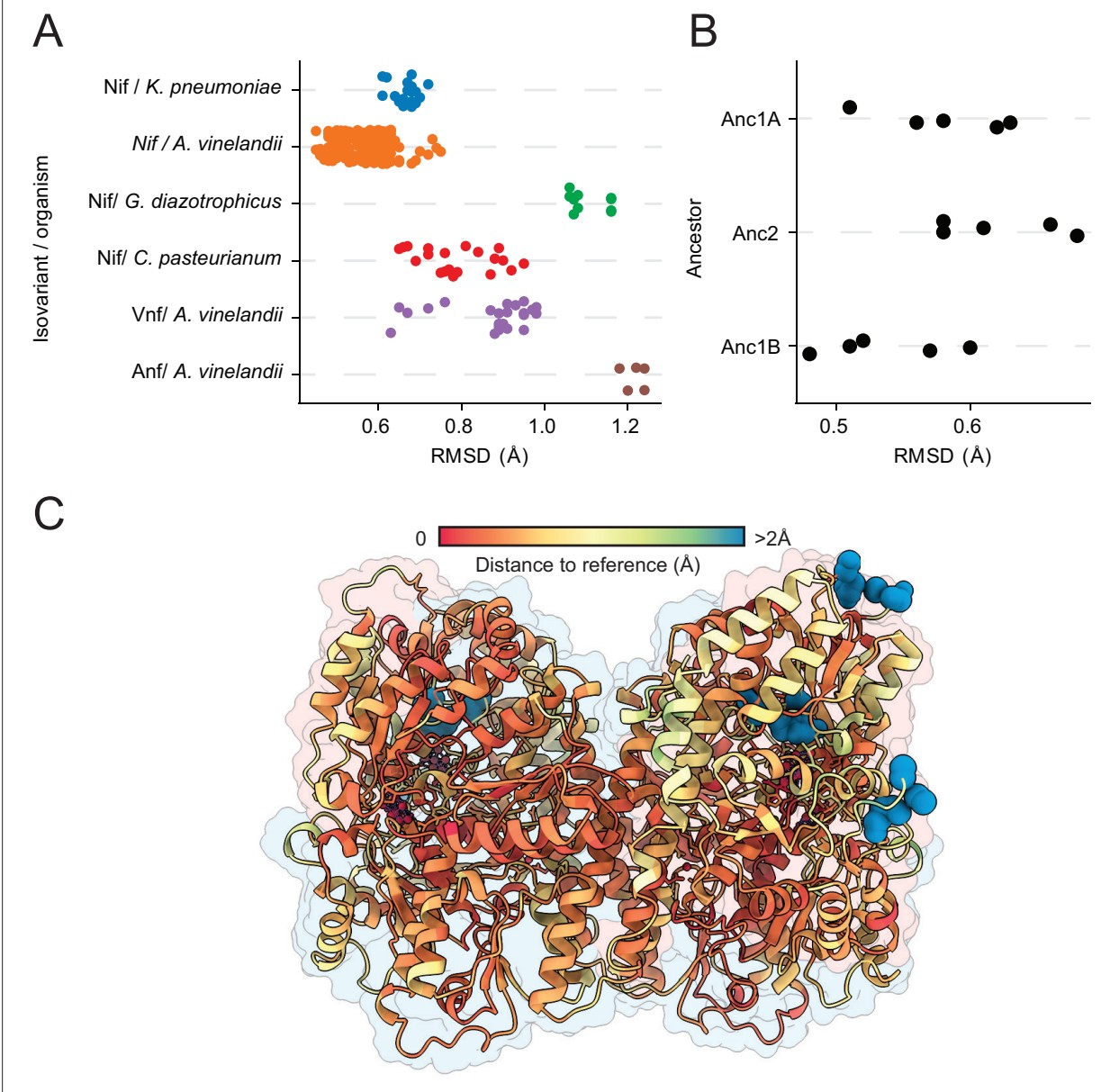

**Figure 6.** Comparison between crystallographic/cryo-EM and AlphaFold-predicted nitrogenase structures. (**A**) Root mean squared deviation (RMSD) between crystallographic/cryo-EM structures available in the Protein Data Bank (PDB) and AlphaFold predictions. Each data point represents a comparison between one of five AlphaFold replicates and each PDB structure. (**B**) RMSD between experimental structures obtained in this work and their AlphaFold predictions. Each data point represents a comparison between one AlphaFold replicate and the experimental structure (**C**) Cα-Cα distance between the Anc1A structural prediction and its crystallographic reference, mapped on the nitrogenase structure through a colormap. Blue residues denote larger distances Cα-Cα distance, which indicate lower prediction fidelity on these regions.

The online version of this article includes the following source data for figure 6:

**Source code 1.** Code to display RMSD differences between experimental and reconstructed nitrogenase structures.

**Source code 2.** Labels for figure display.

**Source data 1.** Alignment distances between experimental and predicted DDKK tetramers using US-align.

three crystallographic and 5378 predicted structures generated in this study represent approximately a 50-fold increase in available nitrogenase complex structures, and importantly includes an extensive sampling of historical structural variation.

Both crystallographic and predicted structure of nitrogenase ancestors highlight the long-term evolutionary conservation of core structural features, consistent with previously characterized

preservation of mechanistic aspects in nitrogenase catalysis (*Garcia et al., 2023*; *Harris et al., 2024*). Nevertheless, historical structural variation sufficiently distinguishes major nitrogenase clades independent of protein sequence information, and this phylogenetic variation is driven primarily by different insertion events that occurred early in nitrogenase evolutionary history. For example, we show that the extension of the K-subunit N-terminus emerged during the early diversification of Group I nitrogenases coinciding with the proliferation of nitrogenases into oxygen-tolerant bacterial lineages and the planetary rise in atmospheric oxygen.

Nitrogenase structural features that might be associated with increased compatibility between the oxygen-sensitive nitrogenase enzyme and oxygen-tolerant host organisms are of significant interest given efforts to transfer N-fixation genes to crops (*Vicente and Dean, 2017*). Furthermore, if such structural features are in fact required for compatibility with oxygen-tolerant hosts, their emergence would represent a pivotal transition in Earth history given the importance of aerobic N-fixers for present-day global nitrogen cycling. For example, *Trichodesmium*, a single genus of oxygenic phototrophic cyanobacteria, today contributes 60–80 Tg of the total 100–200 Tg fixed N produced annually in modern oceans (*Bergman et al., 2013*). It was also proposed that the $O_2$ sensitivity of nitrogenase limited the extent of oxygen production by primary producers (*Mrnjavac et al., 2024*).

The occurrence of the NifK N-terminus extension in oxygen-tolerant diazotrophs, combined with increased inter-subunit interactions and affinity predicted by our in silico analyses, points to a possible functional role in response to the planetary rise of oxygen. Aerobic diazotrophs today utilize a variety of oxygen protection strategies for N-fixation. These include physical separation of nitrogenases into anaerobic, differentiated cells, temporal decoupling of N-fixation from oxygen production in cyanobacteria, and increased rates of aerobic respiration (*Mus et al., 2019*; *Robson and Postgate, 1980*). Further, the Shethna protein II was found to shield nitrogenase from temporary oxygen exposure by forming a complex with nitrogenase proteins and rendering the enzyme inactive yet oxygen-tolerant (*Schlesier et al., 2016*). Nevertheless, evolutionary modifications to the nitrogenase enzyme itself that improve its compatibility in aerobic diazotrophs have not yet been identified. The NifK N-terminus extension discussed here may play a direct role, by stabilizing oxygen-sensitive, intersubunit protein interactions within the nitrogenase enzyme. Alternatively, it might play an indirect role by shaping potential interaction sites with accessory proteins (like the Shethna protein II) that improve oxygen protection. Notably, while the NifK extension may contribute to oxygen protection, it is absent in Vnf and Anf nitrogenases, which are nevertheless present in aerobic and facultative anaerobic diazotrophs. Given the dramatic environmental shifts brought by the GOE, it remains uncertain to what extent these emergent changes were beneficial at the time of their appearance. For example, within the framework of constructively neutral evolution, these changes may have initially been neutral, only later acquiring adaptive value as environmental conditions or biological contexts shifted (*Brunet and Doolittle, 2018*).

Collectively, we identify major structural transitions in the history of nitrogenase enzymes that can serve as a foundation for future experimental studies examining the interplay between cellular and environmental factors that impact enzyme evolution. Nonetheless, the insights gained by structure prediction methods carry inherent uncertainty. For example, detailed active-site side chain prediction would require explicit incorporation of metal-cofactors, a feature not yet included in our AlphaFold2-based pipeline. Future iterations of AlphaFold, with enhanced capacity to capture finer-scale features (e.g., side chain structures), may enable higher-resolution estimates and reveal subtle structural and historical variations.

Our study shows that, while nitrogenase retains a conserved multimeric core structure, it has also evolved novel structural features in alignment with key environmental transitions in Earth's history. Subtle, modular changes, particularly those distal to the active site, likely played a crucial role in the enzyme's adaptation and long-term persistence of over geologic time. These findings suggest that transient regulatory adaptations in response to global shifts may have significantly influenced the trajectory of protein evolution. Reconstructing the lost histories of ancient metabolisms (*Kaçar, 2024*), including their structural transformations, opens new avenues for exploring how climatological and ecological forces shape molecular evolution.

## Materials and methods

### Key resources table

| Reagent type (species) or resource | Designation | Source or reference | Identifiers | Additional information |
|---|---|---|---|---|
| Strain, strain background (*Azotobacter vinelandii*) | DJ | DOI:10.1128/JB. 00504–09 | | Dennis Dean, Virginia Tech; Wild-type (WT); Nif+ |
| Genetic reagent (*A. vinelandii*) | DJ2102 | DOI:10.1016/bs.mie.2018. 10.007 | | Dennis Dean, Virginia Tech; Strep-tagged WT NifD; Nif+ |
| Genetic reagent (*A. vinelandii*) | DJ2278 | Other | | Dennis Dean, Virginia Tech; ΔnifD::KanR; Nif- |
| Genetic reagent (*A. vinelandii*) | AK013 | DOI:10.7554/eLife.85003 | | 'Anc1A'; ΔnifD::nifDAnc1A; Nif+ |
| Genetic reagent (*A. vinelandii*) | AK023 | DOI:https://doi.org/10. 7554/eLife.85003 | | 'Anc1B';ΔnifHDK::nifHDKAnc1B; Nif+ |
| Genetic reagent (*A. vinelandii*) | AK014 | DOI:https://doi.org/10. 7554/eLife.85003 | | 'Anc2';ΔnifD::nifDAnc2; Nif+ |
| Software, algorithm | MAFFT | MAFFT | RRID:SCR_011811 | |
| Software, algorithm | trimAl | trimAl | | |
| Software, algorithm | RAxML | RAxML | RRID:SCR_006086 | |
| Software, algorithm | ModelFinder | ModelFinder | | |
| Software, algorithm | IQ-Tree | IQ-Tree | RRID:SCR_017254 | |
| Software, algorithm | PAML | PAML | RRID:SCR_014932 | |
| Software, algorithm | MOLREP | MOLREP | | |
| Software, algorithm | BUSTER | BUSTER | RRID:SCR_015653 | |
| Software, algorithm | REFMAC5 | REFMAC5 | | |
| Software, algorithm | ColabFold | ColabFold | RRID:SCR_025453 | |
| Software, algorithm | HH-suite | HH-suite | RRID:SCR_016133 | |
| Software, algorithm | US-Align | US-Align | | |
| Software, algorithm | Prody | Prody | | |
| Software, algorithm | FreeSASA | FreeSASA | | |
| Software, algorithm | RING | RING | | |
| Software, algorithm | ChimeraX | ChimeraX | RRID:SCR_015872 | |

## Nitrogenase ancestral sequence reconstruction, selection, and resurrection in *A. vinelandii*

All extant and ancestral nitrogenase protein sequences were drawn from a previous nitrogenase sequence dataset (*Garcia et al., 2023*). Briefly, 385 nitrogenase sequences and 385 outgroup sequences were curated from the NCBI nr database by BLASTp for phylogenetic tree building and ancestral sequence reconstruction. For ancestors Anc1A and Anc2, aligned sequences (MAFFT v7.450; *Katoh and Standley, 2013*) were trimmed (trimAl v1.2; *Capella-Gutiérrez et al., 2009*) and both tree reconstruction and ancestral sequence inference were performed by RAxML v8.2.10 (*Stamatakis, 2014*) with the LG+G+F evolutionary model. Model selection was performed by ModelFinder (*Kalyaanamoorthy et al., 2017*) in IQ-TREE v.1.6.12 (*Nguyen et al., 2015*). For ancestor Anc1B, the untrimmed sequence alignment was used for tree reconstruction by RAxML and ancestral sequence inference by PAML v4.9j (*Yang, 2007*) using the LG+G+F model. This second analysis was conducted due to concerns that RAxML v8.2 does not perform full, marginal ancestral sequence reconstructed as described by *Yang et al., 1995*. Anc1A and Anc1B are equivalent in their set of descendants and, despite the difference in reconstruction methods, have NifD proteins with 95% identity (*Figure 5—figure supplement 3*).

In addition to the ancestral sequences described above, we calculated the statistical uncertainty of ASR by generating five alternative variants for each ancestor in the nitrogenase phylogeny. These

include one variant that contains the second most probable residue at each ambiguous position (probability of most probable ancestral state <0.7) ('*altall*') and another four variants selected by randomly sampling the ASR posterior probability distribution at each ambiguous site ('*alt2*,' '*alt3*,' '*alt4*,' '*alt5*'). The *altall* and *alt2-5* sequences correspond to the alternative sequences considered in *Figure 1*.

Engineering of *A. vinelandii* strains was performed as previously described, replacing the native WT *nifD* gene with an ancestral variant (*Garcia et al., 2023*). A. *vinelandii* WT ('DJ'), DJ2278 (ΔnifD::KanR), and DJ2102 Strep-II-tagged WT NifD were generously provided by Dennis Dean (Virginia Tech) for strain engineering (*Santos, 2018*). Following genomic integration of ancestral nitrogenase genes into *A. vinelandii*, transformants were passaged at least three times to ensure phenotypic stability prior to storage at –80°C in phosphate buffer containing 7% DMSO. Verification of the engineered strains was performed by Sanger sequencing of the *nifD* region as well as whole genome sequencing.

## Expression and purification of nitrogenase proteins *A. vinelandii*

Cells from DMSO stock were directly grown in a pre-culture using liquid Burk medium containing ammonium sulfate at 37°C and 180 rpm until an optical density at 600 nm of at least 1.5 was reached. Further, a second pre-culture without ammonium sulfate was grown under the same conditions and used to inoculate 500 mL main cultures at 30°C and 180 rpm. Cells were harvested when an optical density at 600 nm of 1.7–2.5 was reached.

### Anc1A

Cells were grown in a 10 L Eppendorf BF120 bioreactor at 30°C in Burk medium containing urea as the fixed N source. Dissolved oxygen (DO) was held at 20% using lab air at a constant 10 SLPM and cascade control of agitation (100–500 rpm) and supplemental $O_2$ (0–30%). At an optical density at 600 nm of ~8, the cells were pelleted by centrifugation and resuspended in Burk medium without a fixed N source to induce expression of nitrogenase. Cells were allowed to express for 4 hours with DO held at 10% with the same air flow and cascade control before harvesting.

### Isolation of MoFe proteins

MoFe protein is oxygen sensitive; therefore, all consecutive steps were conducted either in an anoxic chamber (Coy Laboratories) under 95% $N_2$ and 5% $H_2$ atmosphere or using modified Schlenk techniques under $N_2$ flow. Anc1 contains two additional crystallographic forms, Anc1A and Anc1B; the two are the same age, and Anc1A contains an ancestral D-subunit protein complexed with WT H- and K-subunit proteins. Anc1B is by contrast composed of ancestral H-, D-, and K-subunit proteins. They were isolated as follows:

### Anc1B (MoFe and Fe protein)

Cell pellets were resuspended in lysis buffer (50 mM Tris/HCl at pH 7.4, 2.5 mM $Na_2S_2O_4$) and opened at 1500 bar in a SUP Maximator HPL-6 at 2°C under anoxic conditions. The cell-free extract was centrifuged at 87,000 × *g* at 4°C for 60 minutes, filtered through a 0.45 μM syringe filter (Filtropur), and was loaded onto two equilibrated 5 mL HiTrap Q HP columns. After washing with 12.5% elution buffer (50 mM Tris/HCl at pH 7.4, 1000 mM NaCl, 2.5 mM $Na_2S_2O_4$), the amount of elution buffer was increased in 2.5% increments up to 35%, followed by steps at 40 and 50%. MoFe and Fe protein eluted after 300 mM NaCl and 325 mM NaCl, respectively. Step elution with 40% elution buffer was used to concentrate. The proteins were loaded separately onto a size-exclusion chromatography (Superdex S200, GE Healthcare) equilibrated with 20 mM Tris/HCl at pH 7.4, 200 mM NaCl, 2.5 mM $Na_2S_2O_4$, concentrated (Vivaspin 20, 50 or 30 kDa MWCO, Sartorius) and flash-frozen in liquid nitrogen. A second size-exclusion chromatography step (Superdex 200 Increase 10/300 GL) was carried out with MoFe protein, which was subsequently concentrated and flash-frozen in liquid nitrogen.

### Anc1A

Cell pellets were resuspended in lysis buffer (50 mM Tris/HCl pH 7.9, 20% glycerol (v/v), 500 mM NaCl, and 2 mM $Na_2S_2O_4$). Cells were lysed with a Nano DeBee 45–2 High Pressure Homogenizer and then ultra-centrifuged to produce cell-free lysate. Cell-free lysate was loaded onto an equilibrated (50 mM Tris/HCl pH 7.9, 500 mM NaCl, and 2 mM $Na_2S_2O_4$) 20 mL StrepTactinXT 4Flow column. The

column was washed with 2 column volumes of equilibration buffer, followed by elution of the protein in a single fraction with elution buffer (50 mM Tris/HCl pH 8, 150 mM NaCl, 50 mM biotin, and 2 mM $Na_2S_2O_4$). Protein was then loaded on an equilibrated (20 mM Tris/HCl at pH 7.4, 200 mM NaCl, 2.5 mM $Na_2S_2O_4$) size-exclusion chromatography (Superdex 200 Increase 10/300 GL). Subsequently, it was concentrated and flash-frozen in liquid nitrogen.

### Anc2

Cell pellets were resuspended in lysis buffer (20 mM Tris/HCl at pH 7.4, 200 mM NaCl, 2.5 mM $Na_2S_2O_4$) and opened, centrifuged, and filtered following the Anc1B protocol. The cell-free extract was loaded onto a with lysis buffer equilibrated Streptavidin XT 4 Flow column. Through the addition of 50 mM Biotin to the lysis buffer, Anc2 eluted. Size-exclusion chromatography (Superdex S200, GE Healthcare) was equilibrated using lysis buffer and the protein loaded. Protein was concentrated (Vivaspin 20, 50 kDa MWCO, Sartorius) and flash-frozen in liquid nitrogen until further use.

## Crystallization and data collection

Protein crystallization was carried out in an anoxic chamber with less than 5 ppm of $O_2$. Solutions used for crystallization were degassed using modified Schlenk techniques, and crystallization was carried out using the sitting-drop vapor diffusion technique in 96-well plates (Anc1A and Anc1B) (Swissci 96-well two-drop plate, Hampton Research) or 24-well plates (Anc2) (24-well crystallization Cryschem M plates, Hampton Research). Crystals that were used for microseeding were added to 50 μL reservoir solution and a PTFE seed bead (Hampton Research). The crystals were crushed by six times vortexing for 30 seconds and then diluted with reservoir solution. 7.5 mM of $Na_2S_2O_4$ was added to the protein prior to crystallization. 2 μL of 10 mg/mL protein solution (Anc2) were mixed with the same volume of reservoir solution containing 4.5% (w/v) of polyethylene glycol 2000, 3350, 4000 and polyethylene glycol mono methylether 5000, 0.1 M MES/NaOH at pH 6.5, 10% (v/v) of ethylene glycol and 0.15 M Mg acetate. 0.2 μL of 1:100 diluted seed were added.

For Anc1A, 0.3 μL of 9 mg/mL protein solution were mixed with 0.4 μL of reservoir solution containing 0.2 M LiSO₄ × 1 $H_2O$, 0.1 M bis-Tris/HCl at pH 6.5, 25%(w/v) of polyethylene glycol 3350. 0.1 μL of 1:400 diluted seed was added. 0.4 μL of 9 mg/mL protein solution (Anc1B) was mixed with 0.4 μL of reservoir solution containing 0.1 M MES/imidazole at pH 6.5, 12.5% of (w/v) of polyethylene glycol 1000, 3350 and (v/v) 2-methyl-2,4-pentandiol each and 0.02 M of 1,6-hexanediol, 1-butanol, 1,4-butanediol, (RS)–1,2-propanediol, 2-propanol, and 1,3-propanediol, each with addition of 0.1 μL of 1:400 diluted seed. 0.8 μL of 8 mg/mL Anc1B-H was mixed with 0.8 μL of reservoir containing 0.1 M MES/NaOH at pH 6.5, 7.5% of (w/v) polyethylene glycol 6000, 8000, 10,000 each, 10% (vol/vol) isopropanol. 0.2 μL of 1:400 diluted seed stock was added.

Crystals were harvested into nylon loops and flash-frozen in liquid nitrogen. Diffraction data for Anc2 were collected at the Swiss Light Source (Paul Scherrer Institute) on beamline X06SA using an EIGER 16M X detector at an X-ray wavelength of 1.0000 Å. Diffraction data for Anc1A and B were collected at the ESRF on beamline ID30B using an EIGER2 X 9M at an X-ray wavelength of 0.8856 A. Diffraction data for Anc1B-H was collected at the ESRF on beamline ID23-1 using an EIGER2 16M CdTe detector at an X-ray wavelength of 0.8856 A.

## Structure solution and refinement

The crystallographic phase problem was solved using the WT structure of MoFe or Fe protein, respectively, from *A. vinelandii* (PDB:3U7Q, 1G1M). A single solution was obtained for the mutants using MOLREP (Anc1B, Anc1B-NifH) (*Vagin and Teplyakov, 2010*) or PhaserMR (Anc1A, Anc2) (*McCoy et al., 2007*). The obtained model was refined using iterative rounds of BUSTER (Anc1A, Anc2) (*Blanc et al., 2004*) or REFMAC5 (all) (*Murshudov et al., 2011*) and model building in COOT (*Emsley et al., 2010*). *Supplementary files 2–5* show the statistics of the refinement process.

## Ancient and extant protein structure prediction

Structure predictions based on AlphaFold-derived methods require two steps: generating deep sequence alignments and predicting structures based on those alignments (*Jumper et al., 2021*; *Senior et al., 2020*; *Tunyasuvunakool et al., 2021*). Additionally, structural templates can be added to the process, but they are not mandatory. The first step depends on searching large sequence

databases. The optimization of this step requires significant memory resources that are limiting in many cases. ColabFold provides a public MMSeqs2 server (*Steinegger and Söding, 2017*) that generates multiple-sequence alignments tailored to AlphaFold requirements to solve this issue (*Mirdita et al., 2022*). However, the public nature of this server implies that users can only perform a limited number of queries per unit of time. The second step consists of running forward predictions on a deep neural network. This step is usually performed in high-performance graphical processing units (GPU). To predict nitrogenase structures, we have considered a *bona fide* protocol and a recycling protocol.

## Bonafide protocol

In this protocol, we rely on the ColabFold methodology, which consists of using MMSeqs2 to generate a deep sequence alignment covering the whole sequence of the query and running forward predictions with three recycles using the AlphaFold neural network. In this step, we avoided using structural templates and optimizing side chains using the AMBER force field.

## Recycling protocol

To predict variant structures with a high level of sequence identity to another reference sequence, we implemented a different pipeline based on recycling the MSA generated by MMSeqs2. The main reason for this additional step is to avoid blocking a public resource with thousands of similar requests. We realigned the corresponding concatenation of sequences to the MSA generated on the structure prediction of the maximum-likelihood ancestral sequence using MAFFT with its options "−−keep-Length" and "−−addResidue" (*Katoh and Standley, 2013*). Previous conversions from A3M to FASTA format were carried out using HH-suite convert.pl script (*Steinegger et al., 2019*). The result of this re-alignment was converted to A3M again, and the files were modified to include the headers specifying the number of chains for each sequence and the length of each sequence in the concatenated dataset. The rest of the protocol proceeded as the *bona fide* protocol, using three recycles without templates or side-chain optimization.

## Protein structure analysis pipeline

The final dataset includes a total of 5378 structures corresponding to extant and ancestral structures. We utilized semi-automatic methods to analyze the high number of structures. Our pipeline for the analysis of structures includes the following steps. (1) Aligning every structure against a template using US-Align (*Zhang et al., 2022*). For the component-1 complexes (chains DDKK), Vnf nitrogenase (PDB: 5N6Y) was used as a template. For component-2, the NifH dimer was used as a template (PDB: 1G1M), both templates from A. *vinelandii*. (2) Modifying the name of chains to match the reference structure, and (3) extracting the pLDDT, predicted local distance difference test, which is the standard uncertainty predictor of AlphaFold (*Jumper et al., 2021*) from the B-factor column of PDB files and trimming regions of the structure with low prediction confidence.

The similarity of two protein structures is computed using US-Align (*Zhang et al., 2022*), which extends the structural alignment capabilities of the TM-align (*Zhang and Skolnick, 2005*) method to proteins containing multiple chains. US-Align provides two metrics that provide an assessment of the similarity between proteins: first, the RMSD, which measures the mean distance in which a set of atoms deviates from another; and the TM-score, which is a metric bound between 0 (no similarity) and 1 (absolute similarity).

$$RMSD = \sqrt{\frac{1}{N} \sum_{i}^{N} \left( x_i^{ref} - x_i \right)^2}$$

where $N$ is the number of atoms, $x_i$ are the 3D coordinates of the aligned atoms, and $x_i^{ref}$ are the 3D coordinates of the reference atoms. In this work, we compute this function solely based on the coordinates of α-carbons.

$$TMScore = \max \left( \frac{1}{L^{ref}} \sum_{i}^{L} \frac{1}{1 + \left( \frac{d_i}{d_0 \left( L^{ref} \right)} \right)^2} \right)$$

where $L^{ref}$ is the length of the reference protein, $L$ is the number of aligned residues, $d_i$ is the distance between a pair of aligned residues, and $d_0(L^{ref})$ is a normalization term to account for the size of the proteins under alignment. We additionally attempted to correct possible bias connected to the structural alignment size using the RMSD100 metric (*Carugo and Pongor, 2001*), defined as

$$RMSD_{100} = RMSD \cdot \frac{1}{1 + \ln\sqrt{\frac{L}{100}}}$$

where $L$ is the length of the structural alignment.

We have combined different programs' outputs to study the structural features among the different nitrogenases. The list of studied features and the respective programs is summarized in *Supplementary file 6*. For normal-mode-derived properties and geometric properties (e.g., gyration radius), we used ProDy (*Bakan et al., 2011*). For affinity predictions, we used PRODIGY (*Vangone and Bonvin, 2015*). For surface solvent-accessible areas, we used FreeSASA (*Mitternacht, 2016*). Finally, for residue-interaction network properties, we used RING (*Clementel et al., 2022*).

## Nitrogenase structure repository

We implemented a lightweight server using the Streamlit framework in Python providing access and the ability to download to all reconstructed structures using taxon IDs, gene names, or species names available at https://nsdb.bact.wisc.edu.

## Visualization

To visualize quantitative data, we employed the Seaborn Python library. To visualize protein structures, we used ChimeraX-v1.5 (*Goddard et al., 2018*).

## Acknowledgements

BCZ acknowledges the Margarita Salas Postdoctoral Fellowship, founded by the Unión Europea - Next Generation EU (BCZ; UP2021-035). This work was supported by the National Aeronautics and Space Administration (NASA) Interdisciplinary Consortium for Astrobiology Research: Metal Utilization and Selection Across Eons, MUSE ICAR [*80NSSC17K0296*] with additional support from the Human Frontier Science Program (HFSP) [*RGY0072/2021*] (KA), WM Keck Foundation, the Hypothesis Fund and the NASA Exobiology Program [*NNH23ZDA001N*]. OE acknowledges support from the European Research Council (Horizon Europe, AdG 101141673). We thank the Center for High Throughput Computing (CHTC), Dr. Derek Harris for assistance in nitrogenase protein purification, Dr. Janet Newlands for the IT support, the Department of Bacteriology at the University of Wisconsin-Madison for providing computing resources, and the members of the Kaçar laboratory for the valuable feedback.

## Additional information

### Funding

| Funder | Grant reference number | Author |
| --- | --- | --- |
| National Aeronautics and Space Administration | 80NSSC17K0296 | Betül Kaçar |
| Human Frontier Science Program | 10.52044/HFSP. RGY00722021.pc.gr.164263 | Kaustubh Amritkar |

| Funder | Grant reference number | Author |
| --- | --- | --- |
| W. M. Keck Foundation | | Betül Kaçar |
| Hypothesis Fund | | Betül Kaçar |
| National Aeronautics and Space Administration | NNH23ZDA001N | Betül Kaçar |
| European Research Council | 10.3030/101141673 | Oliver Einsle |

The funders had no role in study design, data collection and interpretation, or the decision to submit the work for publication.

## Author contributions

Bruno Cuevas Zuviría, Data curation, Formal analysis, Validation, Investigation, Visualization, Methodology, Writing - original draft, Writing – review and editing; Franka Detemple, Data curation, Formal analysis, Investigation, Visualization, Writing – review and editing; Kaustubh Amritkar, Data curation, Formal analysis, Validation, Investigation, Writing – review and editing; Amanda K Garcia, Data curation, Visualization, Writing – review and editing; Lance Seefeldt, Investigation, Writing – review and editing; Oliver Einsle, Resources, Supervision, Investigation, Writing – review and editing; Betül Kaçar, Conceptualization, Data curation, Formal analysis, Supervision, Funding acquisition, Investigation, Writing - original draft, Project administration, Writing – review and editing

## Author ORCIDs

Bruno Cuevas Zuviría https://orcid.org/0000-0003-1479-9442
Franka Detemple https://orcid.org/0009-0000-0395-7527
Kaustubh Amritkar http://orcid.org/0009-0000-5270-3451
Lance Seefeldt https://orcid.org/0000-0002-6457-9504
Betül Kaçar https://orcid.org/0000-0002-0482-2357

Reviewer #1 (Public review): https://doi.org/10.7554/eLife.105613.4.sa1
Reviewer #2 (Public review): https://doi.org/10.7554/eLife.105613.4.sa2
Author response https://doi.org/10.7554/eLife.105613.4.sa3

# Additional files

## Supplementary files

Supplementary file 1. Crystallographic or cryo-EM nitrogenase structures presently available in the Protein Data Bank (https://www.rcsb.org/), accessed September 2023. H, D, K, and G refer to the respective H, D, K, and G-subunits of nitrogenase. The numbering refers to the oligomeric state for each of the subunit in the complex.

Supplementary file 2. Data collection and refinement statistics for Anc2.

Supplementary file 3. Data collection and refinement statistics for Anc1A.

Supplementary file 4. Data collection and refinement statistics for Anc1B HH.

Supplementary file 5. Data collection and refinement statistics for Anc1B DDKK.

Supplementary file 6. Structural features analyzed in the massive nitrogenase structure prediction, with their respective programs.

MDAR checklist

## Data availability

All codes are available at Github: http://github.com/kacarlab/nif-structures (copy archived at *Cuevas Zuviría, 2025*). All crystal structures are available at the Protein Database: Anc1A DDKK: 9HB9, Anc 1B HH: 9HAZ, Anc1B DDKK: 9HBN and Anc2 DDKK: 9HBC. All structures are available at the nitrogenase structure repository at the UW-Madison Department of Bacteriology hosted https://nsdb.bact.wisc.edu and at Zenodo https://doi.org/10.5281/zenodo.13351063.

The following datasets were generated:

| Author(s) | Year | Dataset title | Dataset URL | Database and Identifier |
| --- | --- | --- | --- | --- |
| Cuevas Zuviría B | 2024 | Nitrogenase Structural DB | https://doi.org/10.5281/zenodo.13351062 | Zenodo, 10.5281/zenodo.13351062 |
| Detemple F, Kacar B, Einsle O | 2024 | *A. vinelandii* nitrogenase MoFe protein Anc1a | https://doi.org/10.2210/pdb9HB9/pdb | Worldwide Protein Data Bank, 10.2210/pdb9HB9/pdb |
| Detemple F, Kacar B, Einsle O | 2024 | *A. vinelandii* nitrogenase Fe protein Anc1b | https://doi.org/10.2210/pdb9HAZ/pdb | Worldwide Protein Data Bank, 10.2210/pdb9HAZ/pdb |
| Detemple F, Kacar B, Einsle O | 2024 | *A. vinelandii* nitrogenase MoFe protein Anc1b | https://doi.org/10.2210/pdb9HBN/pdb | Worldwide Protein Data Bank, 10.2210/pdb9HBN/pdb |
| Detemple F, Kacar B, Einsle O | 2024 | *A. vinelandii* nitrogenase MoFe protein Anc2 | https://doi.org/10.2210/pdb9HBC/pdb | Worldwide Protein Data Bank, 10.2210/pdb9HBC/pdb |

The following previously published dataset was used:

| Author(s) | Year | Dataset title | Dataset URL | Database and Identifier |
| --- | --- | --- | --- | --- |
| Sumalee B, Sirinapa L, Jenjira T, Setasak S | 2011 | A survey method and new design lecture chair for complied ergonomics guideline at Classroom Building 2 Suranaree University of Technology, Thailand | https://doi.org/10.5281/zenodo.1335106 | Zenodo, 10.5281/zenodo.1335106 |

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
