## [Editor Report · eLife Assessment]

This **valuable** study presents computational analyses of over 5000 predicted extant and ancestral nitrogenase structures. The data analyses are **convincing**, it offers unique insights into the relationship between structural evolution and environmental and biological phenotypes. The data generated in this study provide a vast resource that can serve as a starting point for studies of reconstructed and extant nitrogenases.

---

## [Referee Report · Reviewer #1 (Public review)]

This was a clearly written manuscript that did an excellent job summarizing complex data. In this manuscript, Cuevas-Zuviría et al. use protein modeling to generate over 5,000 predicted structures of nitrogenase components, encompassing both extant and ancestral forms across different clades. The study highlights that key insertions define the various Nif groups. The authors also examined the structures of three ancestral nitrogenase variants that had been previously identified and experimentally tested. These ancestral forms were shown in earlier studies to exhibit reduced activity in Azotobacter vinelandii, a model diazotroph.

---

## [Referee Report · Reviewer #2 (Public review)]

Summary:

This work aims to study the evolution of nitrogenanses, understanding how their structure and function adapted to changes in environment, including oxygen levels and changes in metal availability.

The study predicts > 3000 structures of nitrogenases, corresponding to extant, ancestral and alternative ancestral sequences. It is observed that structural variations in the nitrogenases correlate with phylogenetic relationships. The amount of data generated in this study represents a massive and admirable undertaking. The study also provides strong insight into how structural evolution correlates with environmental and biological phenotypes.

---

## [Author Response]

The following is the authors’ response to the previous reviews

**Reviewer #1 (Public review):**
Comments on revisions:I appreciate the authors responding to my comments. I think Fig. S10 helps put the structural data into more context. It would be helpful to make clearer in the legend what proteins are being compared, especially in 10C.Although I can see why the authors focus on the NifK extension and its potential connection to oxygen protection, I would point out that Vnf and Anf do not have this extension in their K subunit, and you find both Vnf and Anf in aerobic and facultative anaerobic diazotrophs. This is a minor point, but I think it is important to mention in the discussion.

We thank the reviewer for their thoughtful comments. We now added an additional line to the Discussion following their recommendation and moved Figure S10 to main text.

**Reviewer #2 (Public review):**
Summary:This work aims to study the evolution of nitrogenanses, understanding how their structure and function adapted to changes in environment, including oxygen levels and changes in metal availability.The study predicts > 3000 structures of nitrogenases, corresponding to extant, ancestral and alternative ancestral sequences. It is observed that structural variations in the nitrogenases correlate with phylogenetic relationships. The amount of data generated in this study represents a massive and admirable undertaking. The study also provides strong insight into how structural evolution correlates with environmental and biological phenotypes.

We thank the reviewer for their summary and positive appraisal.